# Evaluation of a clinical decision rule to guide antibiotic prescription in children with suspected lower respiratory tract infection in The Netherlands: A stepped-wedge cluster randomised trial

Josephine S. van de Maat[1]*, Daphne Peeters[2], Daan Nieboer[3], Anne-Marie van Wermeskerken[4], Frank J. Smit[5], Jeroen G. Noordzij[6], Gerdien Tramper-Stranders[7], Gertjan J. A. Driessen[2], Charlie C. Obihara[8], Jeanine Punt[9], Johan van der Lei[10], Suzanne Polinder[3], Henriette A. Moll[1], Rianne Oostenbrink[1]

1 Department of General Paediatrics, Erasmus MC–Sophia Children's Hospital, Rotterdam, The Netherlands, 2 Department of Paediatrics, HAGA–Juliana Children's Hospital, Den Haag, The Netherlands, 3 Department of Public Health, Erasmus MC, Rotterdam, The Netherlands, 4 Department of Paediatrics, Flevoziekenhuis, Almere, The Netherlands, 5 Department of Paediatrics, Maasstad Ziekenhuis, Rotterdam, The Netherlands, 6 Department of Paediatrics, Reinier de Graaf Gasthuis, Delft, The Netherlands, 7 Department of Paediatrics, Franciscus Gasthuis & Vlietland, Rotterdam, The Netherlands, 8 Department of Paediatrics, Elisabeth–TweeSteden Ziekenhuis, Tilburg, The Netherlands, 9 Department of Paediatrics, LangeLand Ziekenhuis, Zoetermeer, The Netherlands, 10 Department of Medical Informatics, Erasmus MC, Rotterdam, The Netherlands

* j.s.vandemaat@erasmusmc.nl

**Data Availability Statement:** Individual participant data that underlie the results reported in this article

## Abstract

### Background

Optimising the use of antibiotics is a key component of antibiotic stewardship. Respiratory tract infections (RTIs) are the most common reason for antibiotic prescription in children, even though most of these infections in children under 5 years are viral. This study aims to safely reduce antibiotic prescriptions in children under 5 years with suspected lower RTI at the emergency department (ED), by implementing a clinical decision rule.

### Methods and findings

In a stepped-wedge cluster randomised trial, we included children aged 1–60 months presenting with fever and cough or dyspnoea to 8 EDs in The Netherlands. The EDs were of varying sizes, from diverse geographic and demographic regions, and of different hospital types (tertiary versus general). In the pre-intervention phase, children received usual care, according to the Dutch and NICE guidelines for febrile children. During the intervention phase, a validated clinical prediction model (Feverkidstool) including clinical characteristics and C-reactive protein (CRP) was implemented as a decision rule guiding antibiotic prescription. The intervention was that antibiotics were withheld in children with a low or intermediate predicted risk of bacterial pneumonia ($\leq$10%, based on Feverkidstool). Co-primary outcomes were antibiotic prescription rate and strategy failure. Strategy failure was defined

will be made available after de-identification at time of article publication, ending 10 years following article publication. Data will be shared with investigators who provide a methodologically sound proposal, designed to achieve aims in the approved proposal, or for individual participant data meta-analysis. Data are deposited in the repository of Data Archiving and Networked Services (DANS, doi: 10.17026/dans-27a-fj4k). Proposals should be directed to info@dans.knaw.nl; to gain access, data requestors will need to sign a data access agreement.

**Funding:** This study was funded by The Netherlands Organisation for Health Research and Development (ZonMW, grant number 836041001 to JvdM) and Innovatiefonds Zorgverzekeraars (B14-205, dossier 2818 to JvdM). No funding bodies had any role in study design, data collection and analysis, decision to publish, or preparation of the manuscript.

**Competing interests:** The authors have declared that no competing interests exist.

**Abbreviations:** aOR, adjusted odds ratio; CRP, C-reactive protein; ED, emergency department; ICC, intracluster correlation coefficient; IQR, interquartile range; NRT, Netherlands Trial Register; PCT, procalcitonin; RTI, respiratory tract infection.

as secondary antibiotic prescriptions or hospitalisations, persistence of fever or oxygen dependency up to day 7, or complications. Hospitals were randomly allocated to 1 sequence of treatment each, using computer randomisation. The trial could not be blinded. We used multilevel logistic regression to estimate the effect of the intervention, clustered by hospital and adjusted for time period, age, sex, season, ill appearance, and fever duration; predicted risk was included in exploratory analysis. We included 999 children (61% male, median age 17 months [IQR 9 to 30]) between 1 January 2016 and 30 September 2018: 597 during the pre-intervention phase and 402 during the intervention phase. Most children (77%) were referred by a general practitioner, and half of children were hospitalised. Intention-to-treat analyses showed that overall antibiotic prescription was not reduced (30% to 25%, adjusted odds ratio [aOR] 1.07 [95% CI 0.57 to 2.01, $p = 0.75$]); strategy failure reduced from 23% to 16% (aOR 0.53 [95% CI 0.32 to 0.88, $p = 0.01$]). Exploratory analyses showed that the intervention influenced risk groups differently ($p < 0.01$), resulting in a reduction in antibiotic prescriptions in low/intermediate-risk children (17% to 6%; aOR 0.31 [95% CI 0.12 to 0.81, $p = 0.02$]) and a non-significant increase in the high-risk group (47% to 59%; aOR 2.28 [95% CI 0.84 to 6.17, $p = 0.09$]). Two complications occurred during the trial: 1 admission to the intensive care unit during follow-up and 1 pleural empyema at day 10 (both unrelated to the study intervention). Main limitations of the study were missing CRP values in the pre-intervention phase and a prolonged baseline period due to logistical issues, potentially affecting the power of our study.

## Conclusions

In this multicentre ED study, we observed that a clinical decision rule for childhood pneumonia did not reduce overall antibiotic prescription, but that it was non-inferior to usual care. Exploratory analyses showed fewer strategy failures and that fewer antibiotics were prescribed in low/intermediate-risk children, suggesting improved targeting of antibiotics by the decision rule.

## Trial registration

Netherlands Trial Register NTR5326.

---

Author summary

### Why was this study done?

- Symptoms of lower respiratory tract infections (RTIs) are very common in children, and the most common reason for children to receive antibiotics in the emergency department (ED).

- A large number of these antibiotic prescriptions may be unnecessary, because most respiratory infections in children are caused by viruses, which are not susceptible to antibiotic treatment.

- Currently, there is no good test to determine whether an infection is caused by a virus or by bacteria, resulting in over-prescription of antibiotics, which in turn can lead to side effects and antibiotic resistance.

## What did the researchers do and find?

- We introduced a clinical decision rule for children with suspected lower RTI in the ED, advising doctors about the child's risk of having a bacterial infection (based on their symptoms), so that they would not prescribe antibiotics to children who had a low or intermediate risk.

- We found that this decision rule did not reduce antibiotic prescriptions when we looked at all children, but that it was safe to use the rule.

- It seemed that, with this decision rule, a higher number of children responded well to their initial treatment (for example, fewer children needed antibiotics in the week after the ED visit, and more children recovered quickly) and that the antibiotic prescriptions shifted from children at low/intermediate risk towards children at high risk of bacterial infection.

## What do these findings mean?

- We observed that it was safe to use the decision rule for guiding antibiotic treatment in children with a suspected lower RTI in the ED.

- It also seemed that, with the decision rule, antibiotics were more often prescribed to children who actually needed them, leading to better recovery from the disease.

- Limitations of our study were that it took more time than expected to organise the logistics of the trial before introduction of the decision rule, and that not all children received a C-reactive protein blood test in the usual care (control) period of the trial, which may have influenced the power of the study.

## Introduction

Respiratory tract infections (RTIs) are the most common diagnosis in febrile children, and the most common reason for antibiotic prescription in children [1]. In children under 5 years, most lower RTIs are viral [2]. Although mortality caused by lower RTIs has decreased significantly over the past decades (currently 1.7 per 100,000 people in Western Europe) [3], antimicrobial resistance due to unnecessary antibiotic prescription is increasing [4]. High variability in antibiotic prescription in children with RTIs in primary as well as hospital care throughout Europe highlights the need for better targeting of antibiotic prescriptions in this patient group [1,5,6].

One of the main challenges when attempting to safely reduce antibiotic prescriptions for lower RTIs in children is the absence of a gold standard for the diagnosis of bacterial pneumonia. Routine chest X-rays are no longer recommended for the differentiation between bacterial and viral causes, and treatment decisions are mostly based on clinical findings [7,8]. Ongoing

research into new biomarkers has not yet provided a new gold standard for clinical practice in the emergency department (ED) [9–11]. In the absence of a gold standard for diagnosing bacterial pneumonia, we need to improve the clinical detection rate of those children who may benefit most from antibiotic treatment of bacterial pneumonia. Clinical prediction models combining clinical characteristics and biomarkers may improve the identification of children who will benefit from antibiotic treatment for community-acquired pneumonia, but they are not used as decision rules in clinical practice [12,13]. The Feverkidstool is a clinical prediction model combining clinical characteristics and C-reactive protein (CRP) to predict the risk of bacterial pneumonia and other serious bacterial infections in children. The model was derived in the ED setting in The Netherlands, and its diagnostic accuracy has been proven in external validation studies in The Netherlands and the United Kingdom [13,14].

In this study we evaluated the impact of the Feverkidstool on clinical practice, as a last step in the development of a prediction model [15]. We translated the Feverkidstool into a decision rule with pre-specified decision thresholds to guide antibiotic treatment for lower RTIs. The primary objective of this study was to safely reduce antibiotic prescription in children under 5 years with suspected lower RTI at the ED, by withholding antibiotics in children at low or intermediate risk of bacterial pneumonia, as predicted by the Feverkidstool.

## Methods

### Study design

We performed a stepped-wedge cluster randomised trial with sequential implementation of a treatment strategy for antibiotic prescription based on a clinical decision rule (hereafter called 'decision rule') in febrile children with suspected lower RTI in the ED. Randomisation at cluster level was chosen to avoid contamination of the intervention effect to control patients. The stepped-wedge design was preferred as, in general, smaller sample sizes are needed than in conventional cluster randomised trials. Because clusters act as their own controls, the intervention effect can be estimated from both between and within cluster comparisons. The sequential implementation of the intervention was deemed superior to a conventional before–after design, given the incorporation of time effects [16]. We performed the trial in 8 clusters (hospitals) between 1 January 2016 and 30 September 2018 in The Netherlands. By design, the trial consisted of 2 phases: a pre-intervention phase, when usual care was provided, and an intervention phase, wherein care was provided according to our intervention (diagram of trial design in Fig 1). A cluster consisted of 1 hospital that was randomised to 1 sequence of treatment. Each hospital was randomised to 1 sequence of treatment, resulting in 8 sequences. The period when all hospitals still performed usual care (the baseline period) was followed by a roll-out period, during which the hospitals switched sequentially to the intervention (antibiotic prescription guided by the decision rule). At intervals of 4 weeks, hospitals were randomised to start the intervention between 28 August 2017 and 12 March 2018. This timing was chosen to take the seasonality of RTIs into account, as most eligible patients were expected during autumn and winter. Given the short duration of illness, the patients included in the different time periods were different people. The original and the final study protocol are available as S1 Text and S2 Text. The trial was approved by the ethics committee of Erasmus MC (MEC-2014-332) and by the participating hospitals. Written informed consent was obtained from the parents of all participants by the treating physician, in both phases of the trial. In the pre-intervention phase, this consent concerned the use of clinical data and performance of follow-up; in the intervention phase, it also concerned the use of the Feverkidstool to guide treatment decisions. Consent was obtained before calculating the predicted risk of the child. The trial was registered in the Netherlands Trial Register (NTR) (NTR5326). As reported in the NTR, 1

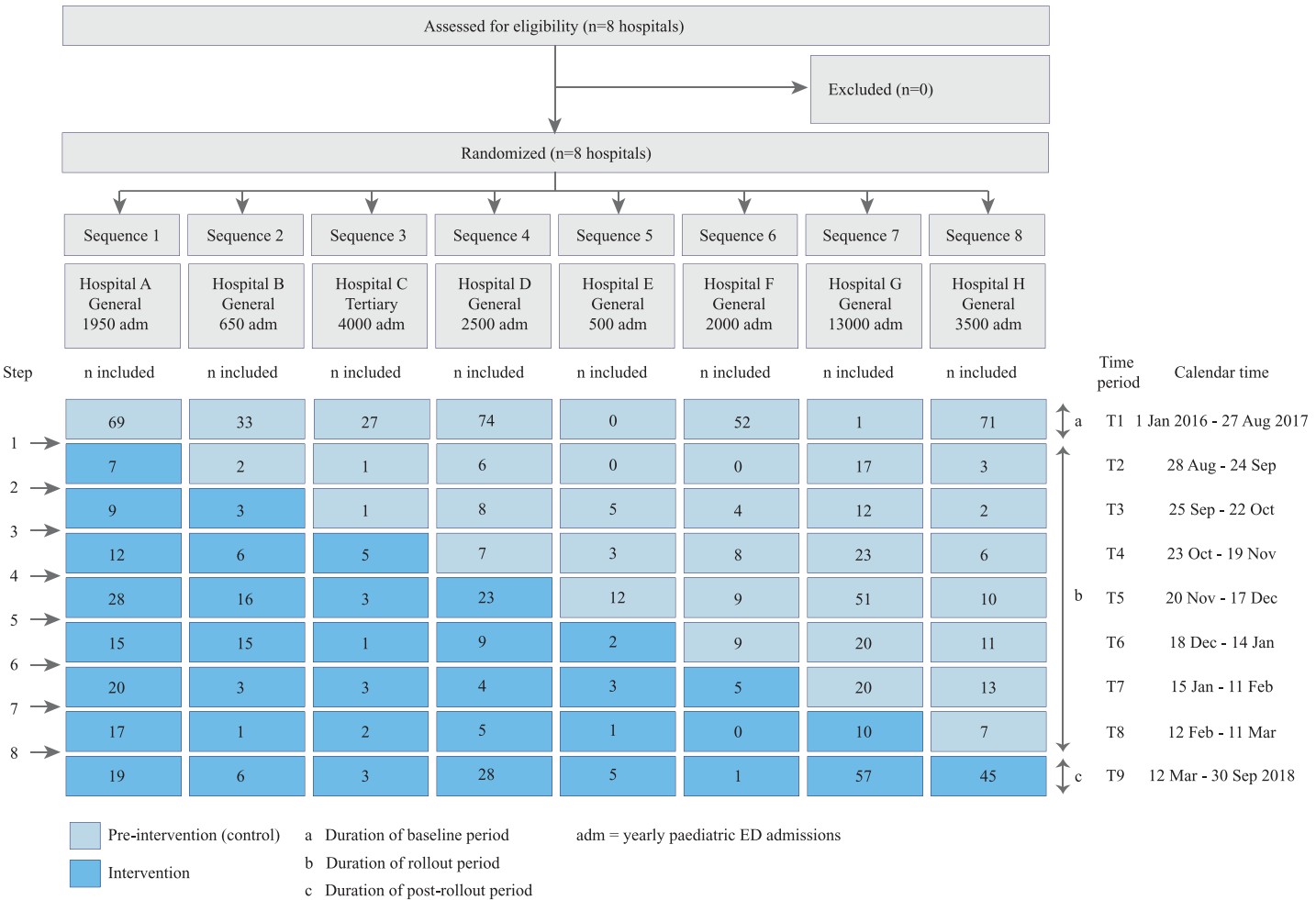

**Fig 1. Design of the trial.** ED, emergency department.

cluster was added during the pre-intervention period, to ensure sufficient inclusions. Interim analysis after the first year of data collection (but before implementation of the intervention) showed substantially higher antibiotic prescription rates than anticipated. Based on the distribution of risks and actual antibiotic prescription rates during the pre-intervention period, the target sample size was adjusted from 1,100 to 900 children, which is also reported in the NTR. No other important methodological changes were made after the start of the trial. The study was reported according to the CONSORT guideline for clinical trials and the extension for stepped-wedge cluster randomised trials (S1 Table).

## Participants

We included children aged 1–60 months that presented with fever (reported by parents or measured as >38.5° C at the ED) and cough or dyspnoea as symptoms of potential lower RTI at the EDs of 8 hospitals in The Netherlands. This target population included children with all different risk profiles, since at presentation in the ED their risk profile was unknown. We excluded children at increased risk of a complicated course: children with relevant comorbidities, antibiotic use in the week prior to ED visit, amoxicillin allergies, another identifiable infectious focus (cutaneous, otitis media, tonsillitis), or signs of complicated pneumonia at

presentation (oxygen saturation < 85%, respiratory insufficiency, empyema, sepsis). Relevant comorbidities were immunodeficiency, congenital heart defect, chronic pulmonary disease, multiple handicaps, and prematurity (born before the gestational age of 32 weeks and aged <1 year at time of presentation). Individual participants were included in the clusters by continuous recruitment by the treating physician in the ED. We included 8 hospitals in 6 cities of the southwest and central area of The Netherlands (a) where paediatricians were responsible for the children presenting at the ED, (b) with varying ED sizes (range 500–13,000 annual paediatric ED admissions), (c) from diverse geographic and demographic regions (inner-city and mixed rural/urban), and (d) of different hospital types (tertiary and general). Hospitals were separated geographically, with no exchange of staff. The hospitals were recruited by the principal investigator (RO).

## Randomisation and blinding

Randomisation of sequences of treatment was performed in July 2017 (after recruitment of all 8 clusters) by a statistician using computer randomisation. The statistician was involved as an advisor in the trial and was based at Erasmus MC. He knew the names of the other participating centres at randomisation, but had no further knowledge of these hospitals or relation to the local researchers. Since 2 hospitals started in August 2017 with the pre-intervention phase due to logistical reasons, these hospitals were randomised to start the intervention after time period 3 (Fig 1). This was accounted for in the original randomisation prior to the rollout period. The trial could not be blinded, because the intervention was the implementation and use of a decision rule by clinicians in the ED, including treatment advice based on the risk score that had to be calculated for each child.

## Intervention

During the pre-intervention phase, all children received usual care. Usual care consisted of triage by a nurse, including the routine measurement of vital signs, followed by a clinical assessment and initiation of therapy by a physician, according to the Dutch and NICE guidelines for febrile children [17,18]. Additional diagnostics were performed at the discretion of the treating physician. CRP testing was often done as part of usual care, but without specific thresholds for decision-making. Other blood tests or chest X-rays were not routinely performed in children with suspected lower RTI, in line with the Dutch guideline, which is based on the British Thoracic Society guideline for the management of children with community-acquired pneumonia [8]. During usual care, antibiotics were prescribed at the discretion of the treating physician. Amoxicillin was usually prescribed as first-line treatment for community-acquired pneumonia [8].

During the intervention phase, a validated clinical prediction model (Feverkidstool) was implemented as a decision rule guiding antibiotic prescription at the cluster level [12,13]. We predefined decision thresholds that would guide antibiotic treatment decisions, balancing positive and negative likelihood ratios and the consequences of over- and undertreatment [12,19]. The intervention was a decision-rule-based treatment strategy for all children with suspected lower RTI in the ED, with a differential effect on risk groups. In children with a low (≤3%) or intermediate (4%–10%) predicted risk of bacterial pneumonia, antibiotics were withheld. In children with a high predicted risk (>10%), usual care was provided, i.e., antibiotics were prescribed at the discretion of the physician. The Feverkidstool included the following predictors: age in years, sex, duration of fever in days, ill appearance (yes/no), chest wall retractions (yes/no), capillary refill time in seconds, hypoxia (oxygen saturation < 94%), tachypnoea (based on Advanced Paediatric Life Support guideline), tachycardia (idem), temperature in degrees

Celsius, and CRP in mg/l. Ill appearance was based on the judgment of the treating clinician. Although ill appearance was not defined by specific criteria, in the development and validation of the Feverkidstool this characteristic appeared to be valid and consistent among different populations [12]. More details about the development of the Feverkidstool have been published previously [12]. The tool was available to all treating physicians as an online digital calculator. The individual predicted risk was calculated after the physician's clinical assessment of the child and CRP testing, but before the treatment decision was made. During both phases of the study, a structured follow-up via telephone was performed 7 days after the ED visit. During the intervention phase, children with an intermediate or high predicted risk received an extra follow-up call 2 days after the ED visit to timely identify potential deterioration of the patient. When children were still hospitalised at those time points, the follow-up information was collected directly from the parents and the patient's electronic health record.

## Outcomes

Primary outcomes were antibiotic prescription at ED discharge (yes/no) and strategy failure within 7 days after the initial ED visit (yes/no). Since the decision rule should not impact patient outcomes negatively (complying with our aim 'to safely reduce antibiotic prescriptions'), we viewed antibiotic prescription and strategy failure as equally important co-primary outcomes. Strategy failure was a composite outcome, based on the follow-up on day 7 and defined as secondary hospitalisation (i.e., hospitalisation during follow-up, after the initial discharge), secondary or switched antibiotic prescription (during follow-up), oxygen dependency or fever up to day 7, or the development of complications. Since there is no single and objective measure of failure of antibiotic treatment strategy, we used this predefined composite outcome for strategy failure. This outcome was chosen to cover different aspects of strategy failure that are important in clinical practice and may be related to the initial treatment strategy at the ED [20]. It includes changes in the treatment strategy for the child (secondary or switched antibiotic treatment and secondary hospitalisation) as well as signs of prolonged or complicated disease (oxygen dependency or fever up to day 7 and complications). Changes in treatment strategy during follow-up were made without specific recommendations in the study protocol. Reasons for switching antibiotic prescription were not systematically recorded. Switching of antibiotics due to adverse drug reaction was considered a strategy failure. We used a short follow-up period of 1 week, assuming that a secondary hospitalisation within this time frame was related to the respiratory illness. All secondary prescriptions and secondary hospitalisations were considered a strategy failure. Secondary outcomes were the level of compliance to the intervention and the number of complications. Compliance was defined as the number of children in whom the Feverkidstool was calculated and who were treated according to the decision rule out of the total number of children included during the intervention phase. Complications were defined as the presence of pleural empyema, parapneumonic effusion (any size), pulmonary abscess, or respiratory insufficiency (need for mechanical ventilation) by day 7. No changes were made to the outcomes after the trial commenced.

## Statistical methods

**Sample size.** We calculated the needed sample size for the 2 co-primary outcomes based on methods by Hussey and Hughes, without accounting for multiple testing [16]. We based our sample size calculation on the complete target population of children with suspected lower RTI in the ED, including all risk groups. Based on previous studies [14], we assumed that 50% of the population would be at low risk, 30% at intermediate risk, and 20% at high risk, with antibiotic prescription rates of 35% (in the low-risk group), 40% (intermediate-risk group),

and 85% (high-risk group). The decision rule was expected to affect risk groups differently: we estimated no difference in antibiotic prescription in the high-risk patients, and a reduction of 10–15 percentage points in the low-risk and intermediate-risk patients, leading to an overall reduction of antibiotic prescriptions of 10 percentage points. The intracluster correlation coefficient (ICC) was unknown, but we assumed that a power of 90% at independency (i.e., no correlation between clusters, ICC of 0) would result in a power of 80% or more in multilevel analysis. We assumed different cluster sizes (small, intermediate, and large clusters) and 3-level seasonal variation in inclusion of patients. All assumptions are listed in S3 Text. Based on these assumptions, we originally estimated a needed sample size of 1,100 children with a suspected lower RTI. Interim analysis of inclusions during the first year showed a higher baseline prescription rate than was assumed. An interim power calculation based on this rate resulted in a needed sample size of 900 children to show superiority of the decision rule for antibiotic prescription with a power of 0.9 and an alpha of 0.05 (see S1 Text). This number was also sensitive to show non-inferiority of the intervention in terms of strategy failure with a non-inferiority margin of 5%: It could detect a 2-fold increase of secondary hospitalisation (the part of strategy failure with available baseline data: 5% at the time of original sample size calculation) with a power of 0.8 and alpha of 0.05. The interim power analysis was performed before introduction of the intervention, so it was blinded to the outcomes of the trial [21].

**Primary analyses.** We used multilevel generalised linear mixed models to calculate the impact of the intervention on our 2 primary outcomes: antibiotic prescription and strategy failure. Hospitals were added as a random effect to take clustering at the hospital level into account. Time period (1–9) was added as a fixed effect to adjust for a secular time trend introduced by the design of the study [22]. In the primary analyses we adjusted for pre-specified factors that may have influenced participation in the study or compliance to the protocol, i.e., age, sex, fever duration, season, and ill appearance. We tested the linearity of the associations between continuous predictors and outcomes. Detailed models can be found in the pre-specified statistical analysis plan (S4 Text). We performed an intention-to-treat analysis, i.e., the intervention population contained all of the children in the intervention phase, including those cases where doctors did not comply to the protocol (Fig 2). We analysed the outcome strategy failure in all children with follow-up information on strategy failure available. We also performed per-protocol analyses to evaluate the impact of the decision rule on the primary outcomes in cases of compliance to the protocol. For this per-protocol analysis, the intervention group consisted only of those children in whom the physicians complied to the protocol (Fig 2).

**Secondary analyses.** We report the level of compliance to the intervention and the number of complications during both phases of the study.

**Pre-specified sensitivity analyses.** We pre-specified 4 sensitivity analyses. First, to estimate the effect of the imputation of missing covariates on our primary analyses, we planned a sensitivity analysis on all covariates with >10% missing values, using different assumptions. Second, to evaluate the effect of loss to follow-up on the outcome strategy failure, we planned a sensitivity analysis assuming that (a) strategy failure occurred in all children with missing follow-up or (b) strategy failure occurred in none of those children. Third, to evaluate the effect of the longer baseline and post-rollout periods, we performed a sensitivity analysis of the primary outcome that only used data from 4 weeks before until 4 weeks after the rollout period (31 July 2017–8 April 2018), resulting in 9 time periods of equal length. Fourth, the level of routine measurement of CRP in the pre-intervention phase differed between hospitals. To adjust for this factor, we performed a sensitivity analysis on the data of only those hospitals that did perform routine CRP measurement in the population throughout both phases of the study.

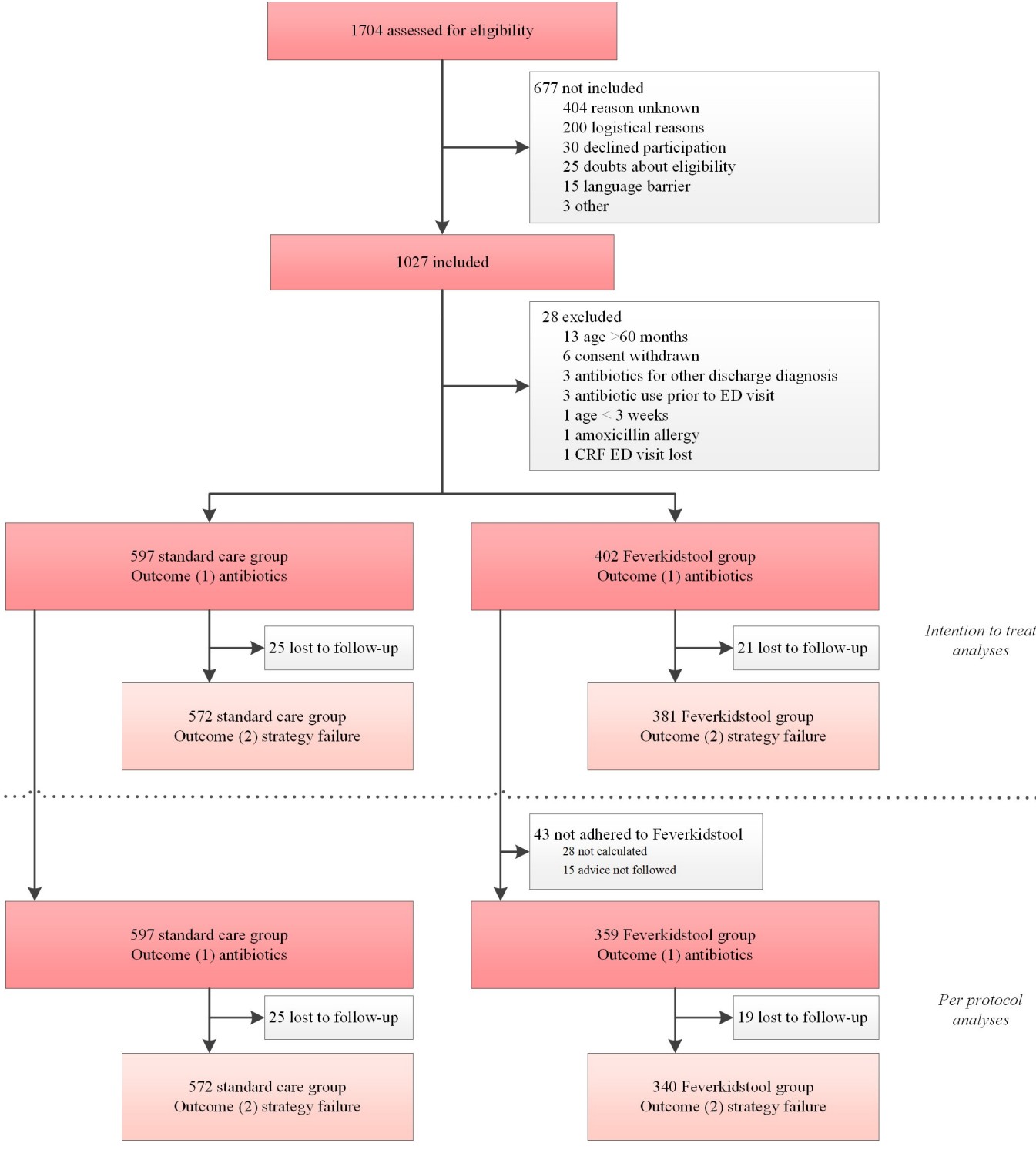

**Fig 2. Flowchart of inclusion.** CRF, case report form; ED, emergency department.

**Exploratory subgroup analysis.** We performed an exploratory subgroup analysis of the primary outcomes in the different risk groups. Because our intervention was to withhold antibiotics in children at low or intermediate risk of bacterial pneumonia, we expected the most

effect on primary outcomes in those risk groups. However, our study was not powered for sub-group analyses, so we performed these post hoc as exploratory analyses, to generate hypotheses for the interpretation of the overall primary results. We analysed the primary outcomes in the low- and intermediate-risk groups combined (≤10%) versus the high predicted risk (>10%) group, testing for a difference in effect using an interaction term (intervention × risk group). For these analyses we used the data of all children in whom the Feverkidstool—and thereby the risk group—could be calculated (complete case analysis), because we could not select sub-groups based on imputed data. We did not perform other post hoc analyses.

**Missing data.** We assumed missing data to be missing at random, and handled missing covariates by means of multiple imputation using the mice package in R (version 3.3.4). The imputation model included all of the variables needed for the primary and sensitivity analyses, as well as additional information on diagnosis, treatment, disposition, and follow-up. Outcome variables (antibiotic prescription and strategy failure) were not imputed, except for the sensitivity analysis evaluating the effect of loss to follow-up on the outcome strategy failure. In this sensitivity analysis we used single imputation only for the outcome variable strategy failure, assuming 100% failure if the variable was missing in one dataset versus 0% failure in another dataset. We did not impute predicted risk for the selection of risk groups in the exploratory analyses. If parents could not be reached for a follow-up on day 7 via telephone, follow-up information was retrieved from the child's electronic health record.

## Results

### Recruitment

The baseline period ran from 1 January 2016 to 27 August 2017, and from 28 August 2017 to 12 March 2018 (the rollout period) the hospitals started the intervention phase one by one every 4 weeks; we collected data until 30 September 2018, when the target sample size was reached (Fig 1). All hospitals adhered to their allocated sequence of treatment and the planned rollout dates. All hospitals that were assessed for eligibility were recruited (*n* = 8); after recruitment, all 8 were randomised to a treatment sequence and included in the analyses (Fig 1). In total, 1,704 children were assessed for eligibility, and 1,027 children included in the trial (375 not included in the pre-intervention phase, 302 in the intervention phase). Of the included children, 28 children met the exclusion criteria, leaving 999 children for analyses of the primary outcome antibiotic prescription. Of these, 46 (5%) were lost to follow-up. Because the outcome strategy failure was based on follow-up, and because we did not impute outcome variables, the remaining 953 children were included in the analyses for strategy failure. Details of patient flow in the trial can be found in Fig 2, and details of inclusion at the cluster level in Fig 1. The main reason for non-inclusion of patients was that the ED was too busy to enrol patients in the trial. The children not included were generally less severely ill, reflected by a lower urgency at triage, fewer antibiotic prescriptions, and fewer hospitalisations (S2 Table).

### Baseline data

The majority of children were male (*n* = 611, 61%), their median age was 17 months (inter-quartile range [IQR] 9 to 30), and most were referred to the ED by a general practitioner (Table 1). One-third of children appeared ill upon ED presentation, and the majority were tachycardic or tachypnoeic or exhibited chest wall retractions. Half of children were hospitalised, for a median duration of 3 days, mainly for oxygen therapy. During the pre-intervention phase, CRP testing was not routinely performed in all children, depending on differences in usual care in the participating hospitals. One hospital was a tertiary care centre; the others

**Table 1. Baseline characteristics of the study population.**

| Characteristic | Pre-intervention<br>*n* = 597 | Intervention<br>*n* = 402 |
|---|---|---|
| **General characteristics** | | |
| Male sex | 364/597 (61%) | 246/402 (61%) |
| Age in months | 17 (9–30) | 17 (9–31) |
| Season | | |
| Spring | 76/597 (13%) | 114/402 (28%) |
| Summer | 55/597 (9%) | 49/402 (12%) |
| Autumn | 198/597 (33%) | 88/402 (22%) |
| Winter | 268/597 (45%) | 151/402 (38%) |
| Way of referral to ED | | |
| General practitioner | 441/578 (76%) | 295/379 (78%) |
| Self | 66/578 (11%) | 45/379 (12%) |
| Other healthcare professional | 71/578 (12%) | 39/379 (10%) |
| Triage level | | |
| Immediate or very urgent | 306/506 (60%) | 182/332 (55%) |
| Urgent | 146/506 (29%) | 121/332 (36%) |
| Standard or non-urgent | 54/506 (11%) | 29/332 (9%) |
| **Signs and symptoms** | | |
| Ill appearance* | 220/572 (38%) | 138/400 (35%) |
| Duration of fever in days | 2 (1–4) | 2 (1–4) |
| Temperature in ˚C | 38.8 (38.1–39.5) | 38.9 (38.1–39.5) |
| Hypoxia (oxygen saturation < 94%) | 144/595 (24%) | 74/401 (18%) |
| Tachycardia | 416/595 (70%) | 274/402 (68%) |
| Tachypnoea | 487/581 (84%) | 315/402 (78%) |
| Retractions | 376/578 (65%) | 237/401 (59%) |
| Dyspnoea | 432/581 (74%) | 290/402 (72%) |
| Wheezing | 233/565 (41%) | 132/395 (33%) |
| Prolonged capillary refill (≥2 seconds) | 96/553 (17%) | 19/401 (5%) |
| **Management** | | |
| C-reactive protein test performed | 372/597 (62%) | 380/402 (95%) |
| C-reactive protein in mg/l | 19 (7–44) | 18 (7–38) |
| Chest X-ray performed | 109/597 (18%) | 49/402 (12%) |
| Discharge diagnosis | | |
| Pneumonia | 204/594 (34%) | 110/401 (27%) |
| Bronchiolitis | 117/594 (20%) | 79/401 (20%) |
| Upper RTI | 197/594 (33%) | 156/401 (39%) |
| Viral induced wheeze | 69/594 (12%) | 49/401 (12%) |
| Other | 7/594 (1%) | 7/401 (2%) |
| Hospitalisation | 329/597 (55%) | 181/402 (45%) |
| Length of stay in days | 3 (2–5) | 3 (2–5) |
| Reason for hospitalisation | | |
| Oxygen therapy | 235/329 (71%) | 132/180 (73%) |
| Intake of antibiotics | 8/329 (2%) | 2/180 (1%) |
| Nebuliser bronchodilator | 10/329 (3%) | 4/180 (2%) |
| Monitoring | 69/329 (21%) | 39/180 (22%) |
| Other | 7/329 (2%) | 3/180 (2%) |
| Type of antibiotic prescribed | | |

(*Continued*)

**Table 1.** (Continued)

| Characteristic | Pre-intervention<br>*n* = 597 | Intervention<br>*n* = 402 |
|---|---|---|
| Amoxicillin | 152/179 (85%) | 84/101 (83%) |
| Amoxicillin/clavulanic acid | 8/179 (4%) | 6/101 (6%) |
| Azithromycin | 17/179 (9%) | 4/101 (4%) |
| Cefuroxime | 2/179 (1%) | 1/101 (1%) |
| Other | 0/179 (0%) | 5/101 (5%) |
| Unknown | 0/179 (0%) | 1/101 (1%) |

Categorical variables are presented as number/total (percentage), and continuous variables as median (interquartile range). The pre-intervention and intervention populations in a stepped-wedge trial cannot be directly compared, but should be adjusted for a secular time trend [22].

*Based on physician's judgment (yes/no).

ED, emergency department; RTI, respiratory tract infection.

were general hospitals (see S3 Table for baseline characteristics per hospital). Annual admissions to the paediatric EDs ranged from 500 to 13,000 (Fig 1).

## Primary and sensitivity analyses

Overall antibiotic prescription was not reduced in the intervention phase (30% versus 25%; adjusted odds ratio [aOR] 1.07, 95% CI 0.57 to 2.01, *p* = 0.75; Table 2). Antibiotic prescription rates per hospital and per time period are provided in S4 Table. Strategy failure decreased from 23% in the pre-intervention phase to 16% in the intervention phase (aOR 0.53, 95% CI 0.32 to 0.88, *p* = 0.01). The per-protocol analysis gave similar results as the intention-to-treat analysis, showing that non-compliance to the decision rule did not influence the observed effect on the primary outcomes. Also the results of the sensitivity analysis with truncated baseline and post-rollout periods were comparable to the analyses on the whole population (Table 2). Two pre-planned sensitivity analyses were not needed: adjusting for missing covariates and adjusting for level of CRP measurement in pre-intervention phase. All covariates for the primary analyses had less than 10% missing values (Table 1), so we assume that no bias was introduced by multiple imputation. There was no difference in the level of CRP measurement between hospitals that performed CRP routinely during the pre-intervention phase and those that did not. Loss to follow-up had no effect on the outcome strategy failure, as shown by the sensitivity analyses that assumed different outcomes for those lost to follow-up (Table 2). Secondary antibiotic prescription was the most frequent reason for strategy failure (Table 2).

## Secondary analyses

In 43/402 (11%) cases, the clinician was not compliant with the decision rule (Table 2). Two complications occurred during the trial: in the pre-intervention phase 1 child was admitted to the intensive care unit during follow-up for mechanical ventilation; in the intervention phase 1 child developed pleural empyema at day 10. Both complications were unrelated to the study intervention, since both patients treated with antibiotics at the first ED visit.

## Exploratory subgroup analysis: Risk groups

We had complete information on all Feverkidstool predictors in 331/597 (55%) of the children in the pre-intervention phase. CRP was the most frequent missing variable (225/597, 38%). The complete case analysis showed that the effect of the decision rule was different across risk

**Table 2. Antibiotic prescription and strategy failure.**

| Analysis and outcome | Number/total (percentage) | | Unadjusted | | Adjusted | |
|---|---|---|---|---|---|---|
| | Pre-intervention | Intervention | OR* (95% CI) | p-Value† | OR‡ (95% CI) | p-Value† |
| **Primary analyses** | | | | | | |
| Intention-to-treat population | | | | | | |
| Antibiotic prescription | 179/597 (30%) | 101/402 (25%) | 1.06 (0.61–1.85) | 0.84 | 1.07 (0.57–2.01) | 0.75 |
| Strategy failure | 131/572 (23%) | 61/381 (16%) | **0.56 (0.34–0.93)** | **0.02** | **0.53 (0.32–0.88)** | **0.01** |
| Per-protocol population | | | | | | |
| Antibiotic prescription | 179/597 (30%) | 83/359 (23%) | 0.89 (0.5–1.61) | 0.71 | 0.96 (0.49–1.88) | 0.92 |
| Strategy failure | 131/572 (23%) | 57/340 (17%) | **0.60 (0.36–1.00)** | **0.05** | **0.56 (0.34–0.93)** | **0.03** |
| **Sensitivity analyses** | | | | | | |
| Truncated baseline and post-rollout periods§ | | | | | | |
| Antibiotic prescription | 66/276 (24%) | 64/279 (23%) | 0.81 (0.45–1.46) | 0.48 | 0.71 (0.38–1.32) | 0.27 |
| Strategy failure | 58/261 (22%) | 46/269 (17%) | **0.57 (0.35–0.94)** | **0.03** | **0.54 (0.33–0.90)** | **0.02** |
| Strategy failure, including missing values | | | | | | |
| Assumption missing = failure | 156/597 (26%) | 82/402 (20%) | **0.56 (0.36–0.88)** | **0.01** | **0.55 (0.35–0.87)** | **0.01** |
| Assumption missing = no failure | 131/597 (22%) | 61/402 (15%) | **0.59 (0.36–0.96)** | **0.03** | **0.56 (0.34–0.91)** | **0.02** |
| **Secondary analyses** | | | | | | |
| Compliance (Feverkidstool calculated and patient treated according to advice) | NA | 359/402 (89%) | | | | |
| Complications¶ | 1/572 (0.1%) | 1/381 (0.2%) | | | | |
| **Strategy failure: reasons** | | | | | | |
| Secondary antibiotic prescription | 45/572 (8%) | 29/381 (8%) | | | | |
| Changed antibiotics during follow-up | 14/572 (2%) | 5/381 (1%) | | | | |
| Secondary hospitalisation** | 16/572 (3%) | 13/381 (3%) | | | | |
| Oxygen need at day 7 | 9/572 (2%) | 1/381 (0.2%) | | | | |
| Fever at day 7 | 47/572 (8%) | 13/381 (3%) | | | | |

Bolding indicates statistical significance.

*Main model: clustered by hospital, adjusted for time period. Time-adjusted intracluster correlation coefficient for antibiotic prescription = 0.04, for strategy failure = 0.

†p-Values based on multivariable logistic regression.

‡Adjusted model: main model further adjusted for age, sex, season, ill appearance, and duration of fever.

§Using data from 4 weeks before until 4 weeks after the rollout period, resulting in 9 time periods of equal length, truncating the prolonged baseline and post-rollout periods.

¶Complications were 1 admission to intensive care unit in the pre-intervention phase and 1 pleural empyema in the intervention phase (both unrelated to study intervention).

**Including 1 admission to the intensive care unit in the pre-intervention group.

NA, not applicable.

groups ($p < 0.01$; Table 3). Antibiotic prescription was lower in the low and intermediate risk groups combined (≤10% predicted risk) during the intervention phase, whereas in the high-risk group prescription rates were higher, but not statistically significantly so. The reduction in strategy failure was observed in the high-risk group (Table 3), mainly via fewer secondary antibiotic prescriptions and less frequent fever at day 7 (S5 Table).

**Table 3. Exploratory subgroup analysis on complete cases (n = 705)\*.**

| Subgroup analysis† | Number/total (percentage) | | Unadjusted | | Adjusted | |
|---|---|---|---|---|---|---|
| | Pre-intervention | Intervention | OR‡ (95% CI) | p-Value§ | OR¶ (95% CI) | p-Value§ |
| **Low/intermediate-risk population (<10%)** | | | | | | |
| Antibiotic prescription | 29/172 (17%) | 15/234 (6%) | **0.37 (0.15–0.94)** | **0.04** | **0.31 (0.12–0.81)** | **0.02** |
| Strategy failure | 29/159 (18%) | 39/218 (18%) | 0.91 (0.43–1.90) | 0.80 | 0.88 (0.42–1.87) | 0.75 |
| **High-risk population (>10%)** | | | | | | |
| Antibiotic prescription | 75/159 (47%) | 83/140 (59%) | 2.04 (0.84–4.94) | 0.11 | 2.28 (0.84–6.17) | 0.09 |
| Strategy failure | 42/155 (27%) | 20/136 (15%) | 0.45 (0.18–1.15) | 0.10 | **0.37 (0.14–0.99)** | **0.05** |

Bolding indicates statistical significance.

\*331/597 (55%) cases were complete in the pre-intervention population, of which 172/331 (52%) were in the low or intermediate risk group (n = 91 low risk; n = 81 intermediate risk); 374/402 (93%) cases were complete in the intervention population, of which 234/374 (63%) were in the low or intermediate risk group (n = 115 low risk; n = 119 intermediate risk).

†Interaction term intervention × risk group p < 0.01.

‡Main model: clustered by hospital, adjusted for time period.

§p-Values based on multivariable logistic regression.

¶Adjusted model: main model further adjusted for age, sex, season, ill appearance, and duration of fever.

## Discussion

We showed that a clinical decision rule did not reduce overall antibiotic prescription in children with suspected lower RTI in the ED, but that it did reduce strategy failure. Exploratory subgroup analyses showed that the intervention influenced the outcomes in the risk groups differently.

Our primary aim was to safely reduce antibiotic prescription in children under 5 years with suspected lower RTI at the ED. We hypothesized that introducing a decision rule as an intervention would safely reduce antibiotic prescriptions in these children. This target population included children with all different risk profiles, since at presentation in the ED their risk was unknown. The first primary endpoint of reducing antibiotic prescription was not met. The other primary endpoint of not increasing strategy failure was met. Moreover, we observed a reduction in strategy failure, suggesting that antibiotic prescriptions were more appropriately targeted to children who benefited from antibiotics. This additional hypothesis was supported by our exploratory subgroup analysis, showing a safe reduction in antibiotic treatments in the low/intermediate-risk group and a (non-significant) increase of prescriptions and a reduction of strategy failures in the high-risk children. This suggests a shift in antibiotic prescriptions from the low/intermediate-risk children towards the high-risk children who had more clinical benefit from the antibiotics. Our power calculation was based on the complete target population of children with suspected lower RTI, assuming a distribution of risk based on previous research. Post hoc sensitivity analysis of the sample size calculation showed that our study was sufficiently powered (power of 0.8), also when accounting for clustering at varying ICC values (range 0.01–0.26) and adjusted for multiple testing. However, we observed a smaller proportion of low/intermediate-risk children in our study population than expected. The shift in antibiotic prescriptions towards high-risk children and the observed smaller proportion of low/intermediate-risk children in our study may explain why we did not detect an overall reduction in antibiotic prescription. However, it must be noted that this finding was based on complete case analysis only and that our study was not powered for subgroup analyses.

We used a composite outcome to define strategy failure. Composite outcomes can be problematic, if the effect of the intervention is mainly driven by less important components [20]. In

our study we found that a reduction in secondary antibiotic prescriptions was the main component of the reduction in strategy failures in the high-risk children and in those in whom we could not calculate the risk score (S5 Table). In low/intermediate-risk children, secondary prescription slightly increased, but without increasing oxygen need or fever at day 7 (proxies for disease severity). There was no increase in complications during the intervention phase. These observations show that our intervention was safe, with reduced strategy failure on clinically important outcomes.

In this trial we used a threshold of 10% to define low/intermediate- versus high-risk patients, based on previous observed diagnostic performance [12], which appeared to be safe. Given the relatively low observed antibiotic prescription rate in the high-risk group, a higher threshold may also be reasonable and more specific, but may carry a risk of increasing strategy failure. These considerations highlight the difficulty in obtaining the optimal balance between reducing overuse of antibiotics (important from a public health perspective) and at the same time striving for the best clinical outcomes for the individual patient [19].

Other impact studies of decision rules for infections in children that combine biomarkers and clinical characteristics are scarce. In a previous impact study, the Feverkidstool was used as a decision rule to guide diagnostic decisions in febrile children in a tertiary hospital. This resulted in a more standardised diagnostic approach, but did not improve the study's secondary patient outcomes, namely antibiotic treatment and hospitalisation [14]. A study of Labscore (a decision rule combining biomarkers) failed to prove its impact on antibiotic prescription in infants with fever without source [23]. Two studies have been reported in non-Western countries on the impact of decision rules on antibiotic prescription [24,25]. A bacterial pneumonia score reduced antibiotic prescription without increasing treatment failure [24], but requires neutrophil testing and a chest X-ray, both of which are not recommended routinely for the management of children with suspected lower RTIs. In Tanzania an algorithm including clinical features, CRP, and procalcitonin (PCT) reduced antibiotic prescription from 94.9% to 11.5% and improved clinical outcomes in febrile children in primary care [25]. Most other studies focused on the impact of single point-of-care biomarkers on antibiotic prescription. A large study in Vietnam showed a reduction of antibiotic use after CRP testing for non-severe acute RTIs in adults as well as in children [26]. In the European ambulatory care setting, there is evidence that CRP testing can reduce immediate antibiotic prescription in children when appropriate guidance is provided to the healthcare professional [27,28]. A randomised controlled trial from Switzerland studied PCT-guided treatment, but found no effect on antibiotic prescription rates [29].

To our knowledge, this is the first multicentre randomised trial designed to measure the impact of a clinical decision rule on antibiotic prescription in children with suspected lower RTI in the ED. A major strength is that our trial studied the impact of a decision rule on usual care. Because the trial was conducted in different settings, mostly general hospitals, we believe our findings are generalisable to general paediatric practice. We had complete information on the outcome antibiotic prescription, good compliance to the protocol, a high follow-up rate, and sufficient power. The sensitivity analyses showed similar results as our primary analyses, confirming the robustness of our findings. There are also some limitations. Logistical problems in starting the trial in 2 hospitals resulted in a longer baseline period before rollout, potentially affecting the power of our study (Fig 1). However, the sensitivity analysis truncating this prolonged baseline period gave results similar to our main analysis, so we believe our overall estimates are valid. Another limitation is the amount of missing Feverkidstool variables in the pre-intervention phase, especially CRP. This did not influence our primary analyses (as CRP was not needed in these models), but limited the number of included patients in the subgroup analyses, where the calculated risk of the Feverkidstool was required. This may have

introduced some bias in the subgroup analyses. Next, not all eligible children could be included in the trial. Doctors in the ED often are under time pressure, leaving insufficient time or attention to recruit patients for a trial, as has been acknowledged by other paediatric ED trials [5,23]. Comparison of the included and non-included children showed that severely ill children were included more frequently. This was the same in both phases of the study, and the rate of eligible children whose families declined participation was also stable over the study phases. Therefore, we believe there was no selection bias introduced by a lack of allocation concealment at the individual level. We believe that we did not miss any children with severe infections, so our results on strategy failure and complications are generalisable.

Although we could not prove an overall reduction of antibiotic prescription, our study implies that guiding antibiotic treatment by a decision rule based on the Feverkidstool is non-inferior in terms of safety in non-complex cases of suspected lower RTI. Moreover, patient outcomes may be improved by better targeting of antibiotics. Implementation of the decision rule in clinical practice would require measuring (point-of-care) CRP, which is not routinely done in all patients with fever and respiratory symptoms [30]. However, we recommend a low threshold for CRP measurements and risk assessments for bacterial pneumonia in these children, and withholding antibiotics in children with a predicted risk of ≤10%, provided that careful safety-netting and good access to healthcare are in place [31]. To avoid the risk of over-prescription in children with a predicted risk of >10%, this approach should be closely monitored. Future research should focus on the safety of higher decision thresholds and on the impact in settings with higher antibiotic prescription rates at baseline, or with a larger proportion of low-risk children. Our observed 30% antibiotic prescription rate at baseline for suspected lower RTIs is lower than what has been described in other European EDs, where antibiotic prescription rates range from 52% to 78% [5,6,13,32]. Even though the populations in many studies cannot be directly compared, a recent paper showed that after adjustment for differences in population, large variability in antibiotic prescription remains [1]. We expect that the effect of our intervention on antibiotic prescription may therefore be larger in settings with a higher baseline prescription rate, or in populations with a larger proportion of low-risk children.

A clinical decision rule for childhood pneumonia did not reduce overall antibiotic prescription, but was non-inferior in terms of strategy failure. Exploratory analyses showed that the intervention reduced antibiotic prescriptions in low/intermediate-risk children, and that it reduced overall strategy failures, suggesting improved targeting of antibiotics by the decision rule.

## Supporting information

**S1 Table. CONSORT checklist.**
(PDF)

**S2 Table. Comparison of included and non-included children.**
(PDF)

**S3 Table. Baseline characteristics per hospital.**
(PDF)

**S4 Table. Antibiotic prescription per hospital and time period.**
(PDF)

**S5 Table. Detailed outcomes per risk group.**
(PDF)

**S1 Text. Final approved trial protocol, version 7 (10 August 2018).**
(PDF)

**S2 Text. Original approved trial protocol version 3 (03 April 2014).**
(PDF)

**S3 Text. List of assumptions for power calculation.**
(PDF)

**S4 Text. Pre-specified statistical analysis plan (14 December 2018).**
(PDF)

## Acknowledgments

We gratefully acknowledge all doctors, nurses, research nurses, and medical students who contributed to the data collection in the participating hospitals. We thank all the included patients and their parents for their willingness to participate in the trial. We acknowledge the Data and Safety Monitoring Board for their support throughout the preparation and conduct of the trial.

## Author Contributions

**Conceptualization:** Johan van der Lei, Suzanne Polinder, Henriette A. Moll, Rianne Oostenbrink.

**Data curation:** Josephine S. van de Maat, Daphne Peeters, Anne-Marie van Wermeskerken, Frank J. Smit, Jeroen G. Noordzij, Gerdien Tramper-Stranders, Gertjan J. A. Driessen, Charlie C. Obihara, Jeanine Punt.

**Formal analysis:** Josephine S. van de Maat, Daan Nieboer.

**Funding acquisition:** Johan van der Lei, Suzanne Polinder, Henriette A. Moll, Rianne Oostenbrink.

**Investigation:** Josephine S. van de Maat.

**Methodology:** Josephine S. van de Maat, Daan Nieboer, Johan van der Lei, Suzanne Polinder, Henriette A. Moll, Rianne Oostenbrink.

**Supervision:** Henriette A. Moll, Rianne Oostenbrink.

**Writing – original draft:** Josephine S. van de Maat.

**Writing – review & editing:** Josephine S. van de Maat, Daphne Peeters, Daan Nieboer, Anne-Marie van Wermeskerken, Frank J. Smit, Jeroen G. Noordzij, Gerdien Tramper-Stranders, Gertjan J. A. Driessen, Charlie C. Obihara, Jeanine Punt, Johan van der Lei, Suzanne Polinder, Henriette A. Moll, Rianne Oostenbrink.

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
