## [Decision Letter · Decision Letter 0]

21 Oct 2019

Dear Dr. van de Maat,

Thank you very much for submitting your manuscript "A clinical decision rule guiding antibiotic prescription for childhood pneumonia: a stepped-wedge, cluster randomized trial" (PMEDICINE-D-19-03060) for consideration at PLOS Medicine. 

[LINK]

In light of these reviews, I am afraid that we will not be able to accept the manuscript for publication in the journal in its current form, but we would like to consider a revised version that addresses the reviewers' and editors' comments. Obviously we cannot make any decision about publication until we have seen the revised manuscript and your response, and we plan to seek re-review by one or more of the reviewers. 

We expect to receive your revised manuscript by Nov 04 2019 11:59PM. Please email us (plosmedicine@plos.org) if you have any questions or concerns.

We look forward to receiving your revised manuscript. 

Sincerely,

Clare Stone, PhD

Managing Editor 

PLOS Medicine

plosmedicine.org

Title – please add country 

Abstract, please add some summary demographic information on the children in the study. Please add p values where 95%Cis are given (also in main text and tables). Please add a sentence on the study’s limitations as the final sentence of the ‘Methods and findings’ section of the abstract. 

Data – data needs to be available (even if through request for restricted data sets because of ethical restrictions) from publication to comply with PLOS data policy. Note also that an author cannot be a point of contact to request access. PLOS Medicine requires that the de-identified data underlying the specific results in a published article be made available, without restrictions on access, in a public repository or as Supporting Information at the time of article publication, provided it is legal and ethical to do so. Please see the policy at 

http://journals.plos.org/plosmedicine/s/data-availability

and FAQs at 

http://journals.plos.org/plosmedicine/s/data-availability#loc-faqs-for-data-policy

Refs need to be in square brackets in the main text and please use the "Vancouver" style for reference formatting, and see our website for other reference guidelines https://journals.plos.org/plosmedicine/s/submission-guidelines#loc-references

The start date for the trial / recruitment in your paper do not match the dates at the trial registration page provided. Please comment. Also there is a discrepancy between the number of registrants needed and how many were recruited (1100 / 900). Please provide more details (line 128) and also comment on whether the study is underpowered. 

Line 451 – please confirm consent was written. 

Please provide a CONSORT checklist as a supp file, using sections and paragraph numbers instead of pages. Apologies if I have missed this. 

Please remove the data statement text from the main Word doc – it is automatically pulled in from the submission form. 

Comments from the reviewers:

Reviewer #1: This articles reports the results of a stepped-wedge trial assessing the impact of an intervention aimed at reducing antibiotics prescription in low-risk children with pneumonia. The trial includes 8 hospitals, sequentially randomised to the intervention. I have listed my comments below:

* I was not able to visualise Figure S1 which was provided in .eps format

* Sample size calculation (cf Lines 197-198): Please provide justification for not adjusting for multiple testing in the presence of two co-primary outcomes. I would also like to understand whether the significant results seen for the strategy failure outcome remain significant using a significance threshold of 2.5% (and a 97.5% confidence interval).

* Please provide the assumptions made behind the statement "The intra cluster correlation coefficient was unknown, but we assumed a power of 90% at independency would result in a power of 80% or more at multilevel analysis." Based on my experience, a loss of only 10% power between an independent and a cluster (stepped-wedge) trial seems somewhat unlikely. I understand that the ICC was unknown but I would like the authors to tell us what ICC was compatible with this minimal loss of power. In addition, given the observed ICC of 0.04, I would like to understand what power was attainable with the sample size and for the targeted difference.

* There appears to be an inconsistency in the OR calculation for Antibiotic prescription. According to my raw calculation 101/402 vs 179/597 leads to an OR of 0.78 in favour of the intervention (or 1.28 using the intervention as the reference and not 1.06 as reported in Table 2). I understand that the unadjusted OR is in fact adjusted for the time period and that the OR estimated from the model is likely to be different from the one obtained via a simple calculation like mine; however, this seems like a big difference. Please confirm.

* The odds ratios for antibiotic prescription and strategy failure (both 'bad' outcomes) appear to be computed in different directions, i.e. once using the intervention period as the reference and once using the pre-intervention period as the reference. This is confusing and I would suggest using the pre-intervention as the reference for all estimates so that an odds ratio smaller than 1 always corresponds to an improvement with the intervention.

* Please consider adding a figure showing the proportion of patients with antibiotic prescription for each period within each hospital so we can visualise the patterns over time.

* Please provide the ICC for both co-primary endpoints in Table 2.

* Please add p-values to Table 2 for at least the primary and secondary analyses.

* In Table 2, please add a note to explain what the sensitivity analysis labelled "population around roll-out period" consist of.

* Looking at the different components of the "strategy failure" outcome, it looks like the main differences are due to fever at Day 7 (and to a lesser extent, oxygen need at Day 7). I am not a clinical expert; however, it seems somewhat implausible to me that a strategy aimed at withdrawing antibiotics would help reduce fever at Day 7. In fact, my understanding was that the risk lied in the opposite direction. I therefore wonder whether this finding (a significant reduction in the proportion of patients experiencing strategy failure) might be due to chance and/or subject to some degree of bias. This needs to be explained/discussed in more details in the discussion.

* Please explain why "the data of patients included during the roll-out period" are thought to "carry most weight in the analyses" (cf lines 363-364).

-Laurent Billot

Reviewer #2: van de Maat and colleagues have submitted a manuscript utilizing a stepped-wedge cluster randomized trial to assess the impact of a clinical decision rule on suspected lower respiratory tract infection in children less than five years of age. Utilizing 999 children enrolled in this study, the decision rule did not impact antibiotic prescribing but did result in less strategy failure. An exploratory analysis among low and intermediate risk children did identify a reduction in antibiotic use and no difference in strategy failure. 

Main concern:

The conclusion was that the clinical decision rule did not reduce overall antibiotic prescription and less strategy failures occurred during the intervention. This is an appropriate conclusion based on the data. However, the authors intervention and power calculation were based on antibiotics not being prescribed in low and intermediate risk children but high risk children were enrolled and included in the analysis. The trial ended up assessing the use of the clinical decision rule in all patients and since likely a significant portion were high risk the impact on antibiotic use was not observed. 

Below are additional thoughts and criticism by section: 

Title: The abstract and manuscript discuss decreasing antibiotics for lower respiratory tract infection and the title states pneumonia. While these are similar, I recommend a change from pneumonia to lower respiratory tract infection.

Abstract: Appropriate

Introduction:

1. The objective of the study was to lower antibiotic use in low or intermediate risk children, however all risk categories were enrolled so the study tested the impact of the clinical decision rule on children presenting with potential LRTI. To address just low or intermediate risk, only these patients should have been enrolled. 

Methods:

1. What number was considered a fever? This was stated in the supplemental protocol. For the readers it would be helpful to have this listed.

2. Page 7 line 136- What kind of otitis? Please be more specific as I'm assuming this is otitis media. 

3. After the intervention was implemented did all patients have the clinical decision rule utilized or only in patients that an informed consent was obtained? When was the informed consent obtained, before or after the use of the calculator?

4. Who performed the informed consent? Clinical coordinator, physician?

5. What constitutes a patient being ill? Was this based on just a clinicians generally belief or were there specific factors that put a patient in the ill category? 

Outcomes

1. Were all antibiotics prescribed considered appropriate? If an antibiotic not recommended for pneumonia was prescribed were these patients excluded?

2. Strategy failure was considered if an antibiotic was switched. If a patient had their antibiotic switched due to allergy or other adverse drug reaction was this considered a failure? Was the reason for switch documented? 

3. Strategy failure was also considered if a secondary hospitalization occurred. I assume this hospitalization had to be due to the same respiratory problem. The authors should specify this as some children could be hospitalized due to an unrelated problem.

4. Complications included parapneumonic effusion or empyema. Was there a certain size of an effusion that had to be present on a chest Xray? Were any effusion on a chest Xray considered a parapneumonic effusion? This should be better specified though with so few complications this point is very minor. 

Power calculation

1. The power calculation was based on low and intermediate risk patients presenting with pneumonia though the study enrolled all patients.

2. The interim analysis changed the needed patients based on more patients in the low risk category receiving antibiotics with the assumption that a greater decrease in antibiotics would be observed. How were the authors able to determine the number of patients in the low risk group if not all patients had the necessary items (eg. crp) to calculate the risk? 

2. The authors mention that this analysis could evaluate non-inferiority in terms of strategy failure. What was the non-inferiority margin that was going to be used to determine if non-inferiority was present? Furthermore, this non-inferiority analysis was not performed so is this even needed to be stated? 

Results:

1. What is the break down in the intervention period of the number of patients in the different risk categories? This could be included in the table and in the pre intervention period just report on those that a score could be calculated. 

2. What is included in "demanding logistics?" Does this include being discharged before the patient could be enrolled since these patients were not as sick as those in the trial. 

3. AS stated before "ill appearance" should be more clearly described since so many children are reported to be ill appearing. I would include this as a footnote in table 1. 

4. In table 1, the pre intervention type of antibiotics added up to 179. Did a patient receive 2? 

5. Table 2- in the complications both pre and post intervention have it stated as 0%. In this situation since there is 1 patient in each group a 0.x% would be appropriate. 

6. What percentage of the patients that would be considered high risk received antibiotics? 

Discussion:

1. The conclusion based on the data is fair but the study ended up including what were likely much more high risk patients in the study since all patient types were included which was not the objective of the study. I do realize a challenge was the lack of knowing the risk group in the pre-intervention phase. 

2. The authors state the lack of CRP did not influence the primary analysis however not being able to compare the risk categories and only do this as an exploratory analysis was not the objective and intent of the study and the intervention. This leads to a conclusion about the decision rule that potentially underestimates the interventions impact. 

3. The authors state that the primary aim was to evaluate the overall impact of a decision rule on antibiotic prescription and strategy failure. The authors stated objective was to "safely reduce antibiotic prescription in children under five suspected of a lower RTI at the ED, by withholding antibiotics in children at low or intermediate risk of bacterial pneumonia, as predicted by the Feverkidstool." The authors should remove first sentence of the section "Interpretation of results…" and focus on what the main objective was. The results were more directed toward the overall impact and not those with just low or intermediate risk.

4. The authors discuss low and intermediate risk patients but present limited data on the number of patients in each category. 

Overall, this manuscript is interesting. The primary objective is not truly evaluated based on the patients enrolled and the results presented. Because of these results, the study did not substantially assess the hypothesis of using the decision rule to decrease antibiotic use, though it appears the decision rule has benefits in both reduction and strategy failures. 

Reviewer #3: This is an important study using a stepped-wedge cluster RCT design to evaluate an intervention to safely reduce antibiotic prescriptions. During the preintervention phase usual care is provided to childhood pneumonia patients in 8 hospitals in The Netherlands. This is followed by an intervention phase during which a validated clinical prediction model (clinical characteristics and C-reactive protein) is used to guide antibiotic prescription in children with uncomplicated pneumonia. The co-primary outcomes in the trial are antibiotic prescription rate and strategy failure comprising secondary antibiotic prescription or hospitalisation, persistent fever or oxygen dependency and complications.

1. The abstract states that the trial could not be blinded and the same is repeated within the text in the method section without further elaboration as to why blinding could not be achieved. This statement on failure to implement blinding can be strengthened in the methods section of the manuscript by explaining why it was not possible to achieve blinding. 

2. The abstract reports that 8 clusters (hospitals) were allocated to sequences of treatment but does not report the number of hospitals randomised to each sequence of treatment and the number of sequences. Similarly, this information is not contained in the body of the manuscript. 

3. The introduction is well written with sufficient detail to justify the conduct of the study. It also contains the objectives for the trial but does not give a rationale for a stepped wedge design. 

4. The authors do not describe their attempts (if any) at allocation concealment. It is noteworthy that allocation concealment can be implemented even in situations where blinding is not feasible and that individual recruitment into cluster trials without concealment of allocation (or blinding) increases the risk of selection bias. 

5. More details are required about the randomisation, recruitment and assignment of clusters to allow the reader ascertain the risk of bias. The authors should provide more details about the statistician who generated the random sequence - independence of the statistician, knowledge of cluster identities, and any other role in the trial. Who recruited the clusters? Who assigned the clusters to the sequence?

6. The trial has two co-primary outcomes: antibiotic prescription and strategy failure (that comprised five components - two of the five components i.e. secondary antibiotic prescription and hospitalisation are based on clinician judgment). The authors refer to antibiotic prescription outcome as a rate in the abstract, while in actual terms this is a proportion and not a rate. The authors overcome some of the difficulties associated with defining and reporting composite outcomes by providing a consistent definition of strategy failure in the abstract, methods and results. The authors also present data for all composite components useful for determining whether a similar effect occurred for all components of the composite. (Cordoba et al. BMJ 2010; 341:c3920 doi:10.1136/bmj.c3920) The following changes could further enhance the reporting of the composite outcome. Providing a rationale for the composite outcome including clinical importance of the components will help with interpretation of the results. At present the discussion and interpretation of the significant findings of reduction in strategy failure seem to ignore the fact that this was a composite outcome and that this outcome was mainly driven by secondary antibiotic prescription. Please refer to Cordoba 2010 on interpreting and discussing effects based on composite indicators.

7. The authors are transparent in reporting the change in sample size calculations after the initial protocol. They also adopt a well described approach for calculating sample size in stepped wedge trials. The sample size calculation as reported is difficult to replicate because details on the number of clusters and clusters allocated at each sequence, cluster size, and ICC or coefficient of variation or assumptions made about these parameters are not presented in the sample size justification. Was an allowance for variation in cluster size made because from supplementary table 6 there was considerable variation in the final cluster sizes? 

8. The sample size justification contains the terms, interim analysis, risk reduction, superiority and non-inferiority. The aim of the study as stated in the introduction section was to safely reduce antibiotic prescription. The sample size calculation should be based on this aim and not to show superiority or non-inferiority as implied in the final sentences in the sample size justification. The authors should consider revising the use of the terms superiority and non-inferiority in their sample size justification, because retaining these terms in the sample size justification will require adopting a different approach to estimating sample size. In the absence of an intracluster correlation coefficient, I found the assumption that a power of 90% for independent data would equate to 80% power for correlated data to be a strong assumption that needs justification.

9. The primary analysis in this paper is based on a multilevel logistic regression model clustered by eight hospitals and adjusting for time as a fixed effect. The correlation of observations in the clusters is accounted for by inclusion of the hospitals as random effects in the multilevel model. This is one of the recommended approaches for analysing intervention effect in stepped wedge studies. The analysis of stepped wedge trials using mixed effect models requires strong assumptions about correlation of observation within each cluster. These assumptions may be inappropriate when the number of clusters is small as is the case in the current study. (Thompson et al. Stats Med. 2018; 37(16) 2487-2500) The authors should explain the potential impact of the few clusters in the trial on the results obtained from analysis approach that is suited for analysis involving a large number of clusters.

10. The sensitivity analysis section does not provide details on the type of imputation that was done for the first of the four sensitivity analyses apart from the statement that covariates with >10% missing data were imputed. The second sensitivity analysis appear to be based on single imputation. The authors should consider commenting on the limitations of their approaches to imputing missing data.

11. The multiple imputation assumed data were missing at random. A sensitivity analysis of departure from missing at random assumption will be helpful in assessing the robustness of the reported results following multiple imputation.

12. The results are appropriately presented based on the analysis approach that was adopted and the recommendation for reporting contained in the CONSORT extension for stepped wedge cluster RCT. Results are presented for both intention-to-treat and per protocol analysis. 

13. The authors state in the discussion "Clinical characteristics of children with missing Feverkidstool variables were comparable to those with complete information. This supports the assumption that missing data were at random, with no major bias to our subgroup analyses". Based on literature on missing data, this is an untestable assumption because the data we would need to test it is missing (Carpenter & Kenward. 2013. Multiple imputation and its applications. John Wiley & Sons, UK)

14. A more detailed discussion of the authors statement that they "included severely ill children more frequently" is warranted because this could point to selection bias.

[LINK]

---

## [Decision Letter · Decision Letter 1]

19 Dec 2019

Dear Dr. van de Maat,

Thank you very much for re-submitting your manuscript "A clinical decision rule guiding antibiotic prescription in children suspected of lower respiratory tract infections in The Netherlands: a stepped-wedge, cluster randomized trial" (PMEDICINE-D-19-03060R1) for review by PLOS Medicine.

I have discussed the paper with my colleagues and it was also seen again by three reviewers. I am pleased to say that provided the remaining editorial and production issues are dealt with we are planning to accept the paper for publication in the journal.

[LINK]

Please also check the guidelines for revised papers at http://journals.plos.org/plosmedicine/s/revising-your-manuscript for any that apply to your paper. If you haven't already, we ask that you provide a short, non-technical Author Summary of your research to make findings accessible to a wide audience that includes both scientists and non-scientists. The Author Summary should immediately follow the Abstract in your revised manuscript. This text is subject to editorial change and should be distinct from the scientific abstract. Please email us (plosmedicine@plos.org) if you have any questions or concerns.

We look forward to receiving the revised manuscript by Dec 23 2019 11:59PM. 

Sincerely,

Louise Gaynor-Brook, MBBS PhD

Associate Editor 

PLOS Medicine

plosmedicine.org

Requests from Editors:

General comments: Please put reference brackets before the full stop (or any punctuation), and after a space at the end of the word/sentence. 

Please revise your title according to PLOS Medicine's style. We suggest "Evaluation of a clinical decision rule to guide antibiotic prescription in children suspected of lower respiratory tract infections in The Netherlands: a stepped-wedge, cluster randomized trial"

Data Availability: Thank you for providing a link to the repository and a point of contact (not an author) for data access. Please explain the reasons for restricting data access to between 12 months and 10 years after publication. PLOS Medicine requires that the de-identified data underlying the specific results in a published article be made available at the time of article publication, provided it is legal and ethical to do so. Please confirm that data will be made available at the time of publication, and not at 12 months following publication. 

Abstract Background: Please expand upon the context of why the study is important. 

Abstract Methods and Findings:

Please include more detail on the setting (e.g. EDs in cities in the Netherlands; types of hospital). 

Line 54 - please clarify what constitutes ‘usual care’ e.g. according to what clinical guidelines

Please provide the number of children included in each group (based on the sequence of treatment available at the ED visited)

Please add a sentence to the abstract to mention the two complications quoted around line 390.

In the last sentence of the Abstract Methods and Findings section, please add “ potentially affecting the power of our study” with relation to the longer baseline period. 

Please begin your Abstract Conclusions with "In this study, we observed ..." or similar.

Line 76 - please revise both instances of ‘less’ to ‘fewer’

Author Summary:

Line 88 - please revise ‘part’ to ‘number’

Line 98 - please revise both instances of ‘less’ to ‘fewer’; please consider another term for ‘failed the initial treatment’

Line 102 - please replace ‘First of all, this means that…’ with ‘We observed that’

Methods:

Please provide more justification in the Methods (as you have in your rebuttal letter) for the adjustment to sample size from 1100 to 900 children.

Line 162 - Please cite the supplementary file containing your CONSORT checklist.

Line 169 - Please include more detail on the setting (e.g. EDs in cities in the Netherlands). 

Line 194 - Please clarify what constitutes ‘usual care’ e.g. according to clinical/local guidelines? 

Line 211 - please define APLS

Line 261 - please amend to Supplementary File S1

Line 337 - please revise ‘less’ to ‘fewer’

Tables 2 & 3 - When a p value is given, please specify the statistical test used to determine it in the legend.

Table S5 - Please specify the statistical test used to determine statistical significance 

Discussion

Please remove all subheadings from within the Discussion i.e. ‘principal findings’ and so on

Please reorganize the Discussion as follows: a short, clear summary of the article's findings; what the study adds to existing research and where and why the results may differ from previous research; strengths and limitations of the study; implications and next steps for research, clinical practice, and/or public policy; one-paragraph conclusion.

Protocol

Supplementary Files 1 and 2 correspond to the final protocol and original protocol respectively. Please amend.

References:

Please ensure all journal titles are appropriately formatted and capitalised e.g. BMJ [ref 20]

CONSORT statement - please format the final column of your checklist as some information appears to have been lost in the .pdf file provided. Please provide section and paragraph numbers. 

Comments from Reviewers:

Reviewer #1: All my comments are addressed.

-Laurent Billot

Reviewer #2: The authors have thoroughly addressed my reviewer comments. I applaud the authors for the detail and time spent in responding to my concerns. I still have 2 thoughts for consideration

1. The authors' state on page 5 line 118, "The absence of a gold standard for bacterial pneumonia impedes deciding upon appropriate treatment." I disagree with this statement. While the true gold standard for diagnosing pneumonia would be sampling the lung, this will never be done for many important reasons. However, the common pathogens to cause community-acquired pneumonia in the age-group studied (eg. Streptococcus pneumoniae) provide us the ability to make determinations of which antibiotics are inappropriate in the population included in this study. The guideline referenced in the response to the reviewers provide recommendations for antibiotics in the outpatient setting. For example, it would be inappropriate to prescribe a quinolone and/or azithromycin. Some experts would suggest prescribing cefdinir for pneumonia would also be inappropriate. Regardless, I think the authors' goal of reducing total antibiotic prescribing is important and they should just note that appropriateness of the antibiotic was not determined. 

2. Is the strategy failure within 7 days of starting or completing the antibiotic? I think it is from starting the antibiotic based on other statements in the manuscript but I think it would be helpful for the reader to have it listed in the outcome paragraph. 

3. The authors should specifically state in the outcomes that changing of antibiotics due to adverse drug reaction was considered a treatment failure. 

4. The authors reported that they did non inferiority but I did not see the non-inferiority margin of 5% listed in the statistical analysis portion of the manuscript.

I believe the effort of these authors in conducting this trial, reporting the results, and addressing the reviewer comments are outstanding. While no study is perfect, I do believe the authors have significantly improved the manuscript and should be published. 

Jason Newland 

Reviewer #3: The authors have responded adequately to most of the comments raised in the review. The following specific comments need to be addressed further:

1) What was the basis for assuming that 90% power under independence equates to 80% power in a scenario of non independence? THis qustion was not addressed in the authors' responses. 

2) Will the figure that contains the range of ICCs used in the sample size calculation (provided in response to reviewer's comments) be included in the final manuscript? If not then it will be useful to write these ranges in the sample size section to allow readers to independently replicate the sample size calculation. I reckon that although the ICC is unknown these ranges were used as plausible estimates. If they are not provided then a reader cannot follow the sample size calculation.

[LINK]

---

## [Editor Report · Decision Letter 2]

6 Jan 2020

Dear Mrs. van de Maat, 

On behalf of my colleagues and the academic editor, Dr. Jason Newland, I am delighted to inform you that your manuscript entitled "Evaluation of a clinical decision rule to guide antibiotic prescription in children suspected of lower respiratory tract infections in The Netherlands: a stepped-wedge, cluster randomized trial" (PMEDICINE-D-19-03060R2) has been accepted for publication in PLOS Medicine. 

PRODUCTION PROCESS

PRESS

PROFILE INFORMATION

Thank you again for submitting the manuscript to PLOS Medicine. We look forward to publishing it. 

Best wishes, 

Louise Gaynor-Brook, MBBS PhD

Associate Editor 

PLOS Medicine

plosmedicine.org